

# Semantic priming and schizotypal personality: reassessing the link between thought disorder and enhanced spreading of semantic activation

Javier Rodríguez-Ferreiro[1], Mari Aguilera[2] and Rob Davies[3]

[1] Grup de Recerca en Cognició i Llenguatge, Departament de Cognició, Desenvolupament i Psicologia de l'Educació, Institut de Neurociències, Universitat de Barcelona, Barcelona, Spain
[2] Grup de Recerca en Cognició i Llenguatge, Departament de Cognició, Desenvolupament i Psicologia de l'Educació, Universitat de Barcelona, Barcelona, Spain
[3] Department of Psychology, Lancaster University, Lancaster, UK

Corresponding author
Javier Rodríguez-Ferreiro,
rodriguezferreiro@ub.edu

## ABSTRACT

The term schizotypy refers to a group of stable personality traits with attributes similar to symptoms of schizophrenia, usually classified in terms of positive, negative or cognitive disorganization symptoms. The observation of increased spreading of semantic activation in individuals with schizotypal traits has led to the hypothesis that thought disorder, one of the characteristics of cognitive disorganization, stems from semantic disturbances. Nevertheless, it is still not clear under which specific circumstances (i.e., automatic or controlled processing, direct or indirect semantic relation) schizotypy affects semantic priming or whether it does affect it at all. We conducted two semantic priming studies with volunteers varying in schizotypy, one with directly related prime-target pairs and another with indirectly related pairs. Our participants completed a lexical decision task with related and unrelated pairs presented at short (250 ms) and long (750 ms) stimulus onset asynchronies (SOAs). Then, they responded to the brief versions of the Schizotypal Personality Questionnaire and the Oxford-Liverpool Inventory of Feelings and Experiences, both of which include measures of cognitive disorganization. Bayesian mixed-effects models indicated expected effects of SOA and semantic relatedness, as well as an interaction between relatedness and directness (greater priming effects for directly related pairs). Even though our analyses demonstrated good sensitivity, we observed no influence of cognitive disorganization over semantic priming. Our study provides no compelling evidence that schizotypal symptoms, specifically those associated with the cognitive disorganization dimension, are rooted in an increased spreading of semantic activation in priming tasks.

## INTRODUCTION

In the influential model proposed by *Meehl (1962)*, schizotypy refers to a personality organization which stems from a pattern of brain functioning that can lead to the development of schizophrenia, given interactions with differences in genetic or social
conditions. Schizotypal personality is characterized by a group of stable personality traits expressed in behaviours similar to symptoms of schizophrenia, encompassing positive (hallucinations, paranoia, magical thinking…), negative (anhedonia, avolition…) and disorganized (odd speech and behavior, thought disorder…) symptoms (*Raine & Benishaw, 1995*) that can be present in the sub-clinical population (see, also, *Ettinger et al., 2014*; *Kwapil & Barrantes-Vidal, 2015*). Thought disorder (cognitive slippage or associative loosening) is one of the fundamental symptoms of schizotypy (*Meehl, 1962*). This characteristic, more specifically referred to as formal thought disorder (as opposed to disorders of thought content, such as delusions), has been proposed to stem from atypical patterns of semantic activation (*Spitzer, 1997*). This hypothesis is based on findings obtained with a range of experimental tasks employed to assess semantic processing, originally, in the study of schizophrenia patients (for a review, see *Doughty & Done, 2009*).

Schizophrenic thinking is often characterized by derailment or loosening of associations between ideas, as well as by obliquely related association or tangentiality (*Rossell, Shapleske & David, 2000*; *Spitzer, 1997*). The intrusion of these oblique or unusual associations into the speech of patients with schizophrenia has been attributed to enhanced distribution of semantic activation during lexical access (*Spitzer, 1997*). This hypothesis is based on a model of semantic memory assuming that words are represented as interconnected nodes which are activated for utterance, and that each time one of the nodes is activated, the activation spreads to related nodes, lowering their threshold for subsequent activation (*Collins & Loftus, 1975*). The problem is that the ways in which differences in schizotypal personality traits are related to differences in semantic processing have been inconsistently observed. We present an investigation comprising two studies designed to clarify the scope of semantic priming, as a marker of enhanced semantic activation, given differences in schizotypal personality traits.

The idea that schizotypal individuals might present enhanced spreading of semantic activation is of ongoing interest because such enhanced activation has been related to possible links between higher levels of schizotypal personality traits and healthy or even favorable functioning, especially in relation to creativity. This hypothesis parallels the possible association between schizophrenia and creativity, which has been suggested to be specifically mediated by the presence of thought disorder (*Barrantes-Vidal, 2004*; *Hasenfus & Magaro, 1976*). In this sense, the generation of uncommon solutions to a given problem by creative individuals could be related to the capacity to activate remote semantic information within the semantic network (*Mohr & Claridge, 2015*). Thus, previous studies have observed increased originality of the responses in semantic fluency tasks (*Kiang & Kutas, 2006*; *Minor & Cohen, 2012*) for volunteers with higher levels of schizotypal traits. These observations can be interpreted to reflect heightened spreading of semantic activation through the semantic system, which could explain the production of infrequent (original) responses in fluency tasks due to the activation of farther nodes in the semantic network.

In the same vein, previous research has shown that individuals with higher scores on schizotypy scales tend to present higher false memory response rates in response to unpresented critical words, having previously been exposed to lists of words semantically

related to them (*Laws & Bhatt, 2005*; *Saunders, Randell & Reed, 2012*). As with the production of more original responses in fluency tasks, increased false memory responses could be argued to result from enhanced spreading of activation from stimulus words to unpresented but semantically related critical words, a transmission of activation that is sufficient to trigger false positives in response to recall or recognition tests. Crucially, in accordance with previous observations with schizophrenia patients (see *Pomarol-Clotet et al., 2008*), some authors argue that this mechanism could also underlie the presence of atypical speech patterns or ideas of reference associating unrelated events in schizotypal individuals (*Mohr et al., 2001*; *Pizzagalli, Lehmann & Brugger, 2001*). Nevertheless, other studies have failed to obtain significant associations between the originality of fluency responses (*Hori et al., 2008*; *Rodríguez-Ferreiro & Aguilera, 2019*) or semantic-based false memory rates (*Corlett et al., 2009*; *Kanemoto et al., 2013*; *Rodríguez-Ferreiro, Aguilera & Davies, 2020*) and differences on schizotypal traits, so the reliability of these observations is unclear.

Several studies have focused on the possible relation between schizotypal personality and semantic priming. As noted by *Spitzer (1997)*, semantic priming might be the most straightforward measure of enhanced (faster and further-reaching) spreading of semantic activation in relation to thought disorder in schizophrenia. The semantic priming effect (*Lucas, 2000*; *Meyer & Schvaneveldt, 1971*) refers to the tendency to respond faster when a target word (e.g., bread) is preceded by a semantically related prime (e.g., butter), compared to an unrelated prime (e.g., doctor) during a speeded language task such as lexical decision (i.e., deciding whether a given letter-string corresponds to a real word or not). This paradigm is relevant for the study of semantic processing because it is assumed to inform us about the process of spreading of semantic activation between related nodes in the semantic system: responses to target words presented after related primes are faster because the semantic activation of the prime has spread through the system, pre-activating related words such as the target (*Yap, Hutchinson & Tan, 2017*).

In a meta-analysis of the results from 36 studies of semantic priming in schizophrenia patients, *Pomarol-Clotet et al. (2008)* found qualified evidence for an increase in semantic priming in patients with thought disorder compared to non-schizophrenic controls. However, the specificity of priming differences was found to be unclear, with patients with thought disorder showing no increase in priming compared to patients without thought disorder. In addition, the meta-analysis could not exclude the possibility that increased semantic priming in patients was an artefact of a general slowing of reaction times in schizophrenia. These qualifications in the clarity of the evidence warrant further investigation of the nature of semantic priming in schizophrenia. A review of the literature also reveals a lack of clarity in otherwise intriguing results from research on the association between differences in schizotypal personality traits and differences in semantic priming.

In Table 1 we summarize the findings of seven studies which have explored the possible relation between schizotypal personality and semantic priming effects. Depending on their aims and initial hypotheses, each study has applied a slightly different experimental approach. Researchers have either used comparisons between priming effects observed in

**Table 1 Summary of previous studies.**

| Study | Questionnaire | n | Comparison | Direct Relation | | Indirect Relation | |
|---|---|---|---|---|---|---|---|
| | | | | Short SOA | Long SOA | Short SOA | Long SOA |
| *Moritz et al. (1999)* | SPQ | 156 | Median split | Null | Null | Null | Null |
| | Frankfurt | 156 | Median split | Positive | Positive | Positive | Null |
| *Morgan, Bedford & Rossell (2006)* | O-LIFE | 58 | High vs. low deciles | Null | Null | NI | NI |
| *Johnston, Rossell & Gleeson (2008)* | O-LIFEDis | 54 | Correlational | Null | Null | Positive | Null |
| *Kiang, Prugh & Kutas (2010)* | SPQ | 28 | Correlational | NA | NA | NA | NA |
| *Neill, Rossell & Kordzadze (2014)* | O-LIFEDis | 36 | High vs. low thirds | Positive | NI | Negative | NI |
| *Rossell et al. (2014)* | O-LIFEDis | 82 | Correlational | Null | Null | Null | Positive |
| *Tan & Rossell (2017)* | O-LIFEDis | 60 | Correlational | Null | Null | NI | NI |

**Note:**
SPQ, Schizotypal Personality Questionnaire (*Raine, 1991*); O-LIFE, Oxford Life Inventory of Feelings and Experiences (*Mason & Claridge, 2006*); O-LIFEDis, disorganized subscale of the O-LIFE; Frankfurt, language subscale of the Frankfurt Complaint Questionnaire (*Süllwold, 1991*); positive: higher scores showed greater priming effects than lower scorers; negative: higher scorers showed smaller priming effects than lower scorers; null: no significant differences between higher and lower scorers or correlation between schizotypy and priming effect; NI: the condition was not included in the study; NA: reaction time data is not available for the study.

high and low scorers on schizotypal personality scales, or have analysed correlations between schizotypy scores and priming effects. The most frequent tools applied to the assessment of schizotypal traits are the Schizotypal Personality Questionnaire (SPQ, *Raine, 1991*) and the Oxford Life Inventory of Feelings and Experiences (O-LIFE, *Mason & Claridge, 2006*). The SPQ was designed to assess the DSM-III-R symptoms of Schizotypal Personality Disorder (*American Psychiatric Association, 1987*) and comprises nine subscales loading onto three factors: cognitive-perceptual or positive schizotypy (unusual perceptual experiences, ideas of reference, magical thinking and paranoid ideation); interpersonal or negative schizotypy (social anxiety, no close friends, constricted affect and paranoid ideation); and disorganization (odd behavior and appearance, odd speech). In contrast, the O-LIFE distinguishes between four dimensions: unusual experiences (positive dimension: perceptual aberrations, magical thinking and hallucinations); introvertive anhedonia (negative dimension); cognitive disorganization (disorganized dimension: poor attention, poor decision making, social anxiety and disorganized speech); and impulsive nonconformity, which refers to unstable mood, antisocial behavior and lack of control. The relevance of this last dimension to the schizotypy construct has been disputed and it is not included in questionnaires based on three-factor solutions of the schizotypy construct like the SPQ. The scores obtained in the disorganized dimensions of these questionnaires have been interpreted to reflect the presence of thought disorder in volunteers (*Johnston, Rossell & Gleeson, 2008*; *Neill, Rossell & Kordzadze, 2014*; *Rossell et al., 2014*). Although it could be argued that the disorganization dimension of SPQ focuses more on thought disorder than the analogous cognitive disorganization dimension of O-LIFE, this second questionnaire has been used more often in the literature studying the relation between schizotypy and semantic processing.

One key aspect of these studies was the examination of whether the association between thought disorder and semantic memory relies on automatic processes (i.e., depending on automatic spreading of semantic activation as suggested by *Spitzer (1997)*) or on

controlled processes. In relation to this second possibility, some authors have argued that language disturbances in individuals with schizophrenia are associated with attentionally-driven (controlled) processes (*Callaway & Naghdi, 1982*). These processes (see *Neely & Keefe, 1989* for an extended discussion) include expectancy effects (i.e., pre-lexical generation of potential targets after the presentation of the prime) and semantic matching effects (i.e., retrospective matching of the target with the prime due to their semantic relation). In order to distinguish between automatic and controlled processes, researchers have manipulated the stimulus onset asynchrony (SOA) between primes and targets. Thus, it is assumed that priming effects obtained with short SOAs, around 250 ms, reflect automatic processing, whereas effects obtained with longer SOAs, longer than 700 ms, are indicative of controlled processing (*Morgan, Bedford & Rossell, 2006*).

A second critical variation among studies has involved the manipulation of the semantic distance between primes and targets. Participants can either be presented with directly related (e.g., lion–tiger) or indirectly related stimulus pairs (e.g., lion–stripes, related through tiger). Following the *Spitzer (1997)* model, and given the association between thought disorder and schizotypy, if thought disorder is rooted in enhanced spreading of activation in the semantic system then we should expect that individuals obtaining high scores on schizotypal personality scales should show greater direct semantic priming. The impact of individual differences on schizotypal scales, especially, on disorganization, should be still more prominent in indirect semantic priming because the presence of faster and further-reaching activation spreading among higher scoring individuals should facilitate the connection between indirectly related pairs of words relative to lower scoring individuals.

In regard to direct semantic priming, *Moritz et al. (1999)* observed greater effects of prime-target relatedness for individuals who scored high on the language subscale of the Frankfurt Complaint Questionnaire (*Süllwold, 1991*), given directly related primes both at short and at long SOAs. This result was partially replicated by (*Neill, Rossell & Kordzadze, 2014*) in a study assessing only short SOA conditions: they found that individuals scoring high on disorganization showed increased direct priming. In contrast, *Moritz et al. (1999)* found no significant differences in priming effects when comparing participants distinguished according to SPQ scores. Further, in studies recording either global O-LIFE scores (*Morgan, Bedford & Rossell, 2006*) or scores on the O-LIFE disorganized subscale (*Johnston, Rossell & Gleeson, 2008*; *Rossell et al., 2014*; *Tan & Rossell, 2017*), no correlation was observed between direct priming effects and individual differences in schizotypy traits.

In regard to indirect semantic priming, *Moritz et al. (1999)* observed greater priming effects for higher scorers on the language subscale of the Frankfurt Complaint Questionnaire (*Süllwold, 1991*) at short SOA but not at long SOA (As noted, they reported no differences in priming when comparing individuals varying on the SPQ). *Johnston, Rossell & Gleeson (2008)* reported a significant positive correlation between scores on O-LIFE cognitive disorganization and indirect priming effects at short SOA. However, in contrast, *Neill, Rossell & Kordzadze (2014)* observed the opposite: participants who scored high on O-LIFE disorganization showed reduced—not increased—indirect priming

effects. Finally, *Rossell et al. (2014)* observed a significant positive correlation between scores in the O-LIFE disorganized dimension and indirect priming effects at long SOA.

Although they do not derive from a reaction time study, we should comment also on the findings reported by *Kiang, Prugh & Kutas (2010)*. Their aim was to assess the possible influence of schizotypal traits on the amplitude of the N400 component, a relatively negative electrocortical component which peaks approximately 400 ms after the appearance of meaningful stimuli like words or pictures. Global SPQ scores were negatively correlated with indirect priming effects on the N400 signal at both short and long SOAs (where priming was observed as differences in N400 amplitudes). Scores on the positive dimension were negatively correlated with direct priming effects at both SOAs, and with indirect priming effects at the shorter SOA only.

These studies present evidence that there are, potentially, associations between differences in schizotypal personality and differences in semantic activation, as reflected in semantic priming effects. However, it is unclear which dimensions of schizotypal personality correlate to semantic priming, and whether a relation between differences in schizotypal personality traits and in priming effects are evident for directly or indirectly related prime-target stimuli, at short or at long SOAs. This lack of clarity in the evidence base is important because the potential association between differences in schizotypal personality and differences in semantic activation is critical to theoretical accounts of schizotypy-related behaviours observed in the sub-clinical population, based in the assumption of enhanced spreading activation. Moreover, it is important because there is also suggestive evidence (also qualified) which relates differences in schizophrenia to differences in semantic priming, and this evidence has similar theoretical implications. What our review of studies on schizotypal personality and priming shows is that further investigation is required to examine trait differences comprehensively across measures of the dimensions of schizotypal personality, and to examine how such trait differences relate to variation in semantic priming under different conditions of relatedness and SOA. We report such an investigation.

While the results of these previous studies are interesting and, of course, have important implications, a close review of the methods suggest strong grounds for expecting that some of the inconsistency among findings may stem from the influence of random variation in samples. This is because the methodological issues that concern us will tend to reduce the sensitivity or precision of analyses and such limitations will tend to afford more space for random variation to have an impact.

Our first concern relates to the selection of sub-groups from participant samples (*Morgan, Bedford & Rossell, 2006*; *Moritz et al., 1999*; *Neill, Rossell & Kordzadze, 2014*) which implies dichotomizing a continuous variable (individual differences on a schizotypy dimension). The problem with this approach is that it has been shown that the practice results in underestimates of effect sizes, reducing the sensitivity or power of statistical hypothesis tests (*Cohen, 1983*). We note, also, that there has been a tendency for studies in the field to involve relatively small participant samples (about 35–60 participants) with few exceptions (*Moritz et al., 1999*), again limiting the sensitivity of statistical tests. We would argue that what is required is an investigation that directly examines the ways in
which individual differences in schizotypy, across a reasonably sized participant sample, interact with indicators of semantic processing.

A further, potentially critical, methodological limitation is the use of difference scores, calculated as an index of semantic processing per individual, to examine the relationship between semantic processing and variation on schizotypal dimensions. Researchers have, for example, calculated the difference between the mean RT of responses under unrelated compared to related priming conditions, calculating the difference score in order to gauge the priming effect per person (*Johnston, Rossell & Gleeson, 2008*; *Morgan, Bedford & Rossell, 2006*; *Moritz et al., 1999*; *Neill, Rossell & Kordzadze, 2014*; *Tan & Rossell, 2017*), or event-related potentials signal amplitude per condition (*Kiang, Prugh & Kutas, 2010*) so that it is then possible to analyse the correlation between the priming effect difference scores and scores on schizotypy scales. The results of such analyses will have important limitations because within-subjects difference scores are notoriously unreliable (as discussed, for example, by *Kliegl, Masson & Richter, 2010*). This unreliability can be understood to result from the implication of the variance sum law (*Howell, 2016*): the variance of the difference of two variables is the sum of their variances (minus the correlation between the variables). This tells us that the variance of the difference score will, in effect, equal the sum of the variance of the mean RT in one condition plus the variance of the mean RT in the other condition. Given that variance about the mean indicates the reliability of estimates so that means associated with wider variances are known to be less reliable, the reduced reliability will limit the sensitivity of analyses. This is because the reliabilities of two measures provide an upper bound on the possible correlation that can be observed between the two measures (*Vul et al., 2009*).

The key problem for researchers seeking to draw conclusions from previous observations is that when power (or sensitivity) is low, studies that show statistical significance will tend to be at risk of presenting exaggerated estimates. *Vasishth et al. (2018b)* present analyses which demonstrate that in low-power scenarios the estimates from repeated samples will fluctuate around the true value, potentially giving rise to errors both in the estimation of the magnitude of the effect and in the estimation of the sign of the effect (*Gelman & Carlin, 2014*). The overestimates will occur because the standard error is relatively large in low-power situations. The consequence of broadly spread sampling distributions of the mean (wider because of larger standard errors) is that there is then a greater chance that extreme values are observed. In the present investigation, we eschewed the use of traditional null hypothesis significance testing (we adopt Bayesian methods) in part because of these problems but also to focus our observations on the estimated magnitude and sign of the effects of interest.

An alternate way of looking at the analysis problem is that we must address it as efficiently as possible with respect to the use of the data collected in test samples. We are seeking to identify the ways in which semantic priming may vary in association with individual differences in schizotypal dimensions. Calculating difference scores per person corresponds to no-pooling analyses in which the per-person difference score identifies the priming for that person, this leaves us prone to using less reliable estimates of priming (based on the data of just that person) and, critically, it ignores what we can find out about
other participants (i.e., the average priming effect across the participant sample). Of course, individuals will vary so we cannot, either, seek to examine the correlation between the priming effect over all participants and variation in schizotypy scores (the complete-pooling approach) because that would ignore the individual differences that can be presumed to be present in the data. What is required is a method of analysis that allows us to estimate the average (or population-level) effects of experimental factors like semantic relatedness, while taking into account the random as well as the systematic variation between individuals tested in our study. That method is provided in multilevel or mixed-effects modelling (*Gelman & Hill, 2007*), and that is the method we use in our analyses.

How, then, can we examine how semantic processing varies among individuals varying in schizotypy? Our approach is to consider the impact of interactions between the effects of semantic prime-target relatedness, or the directness of relatedness, and the effect of variation in schizotypal dimension. If the question is: how do individuals varying on a schizotypal dimension vary in how they process semantics?—we can frame the means to address it by supposing that semantic processing can be gauged in terms of the effect of relatedness, the average difference between related and unrelated primes. If we are seeking to test how that difference is, itself, different among individuals at different levels of schizotypy then we are, by definition, seeking to test the interaction between the effect of prime-target relatedness and the effect of schizotypy (*Cohen et al., 2003*) because we are examining how the effect of one variable (relatedness) can vary at different levels of a second variable (schizotypy). This is the approach we take in our investigation.

All in all, although the idea that schizotypal personality is associated with semantic disturbances is commonly espoused, it remains unclear under which specific circumstances schizotypy affects semantic priming or whether it does affect it at all. For this reason, some authors have stressed the need to replicate previous studies in order to clarify the nature of the association between schizotypy and priming effects (*Rossell et al., 2014*). We present the results of two studies assessing the association between schizotypy and semantic processing. Given the focus of previous studies on thought disorder, we hypothesized, firstly, that cognitive disorganization would be positively related to semantic priming, with volunteers scoring high on this dimension showing greater priming effects than those scoring low. We focus on the cognitive disorganization dimension of schizotypy because it is assumed to be the most closely related with thought disorder (*Mason & Claridge, 2006*; *Raine, 1991*). We predicted that, on average, semantic priming would be observed as a difference in response latencies such that responses were faster to word targets with related primes than to targets with unrelated primes. However, we predicted an interaction such that this related vs. unrelated difference would be larger for participants with higher scores on cognitive disorganization scales. Although, as we have already mentioned, the disorganized dimension of the SPQ might be more adequate to assess thought disorder than the corresponding dimension of the O-LIFE, we included versions of both scales in our study in order to facilitate comparisons with previous research.

We aimed to test, secondly, whether the association between schizotypal personality and semantic memory relies on processes that are automatic (i.e., enhanced automatic spreading of semantic activation) or controlled (e.g., attentionally-driven expectancy or matching effects). We hypothesized that if variation between individuals in semantic priming reflected the impact of schizotypy on either automatic or controlled semantic processes then we should expect to observe an interaction between the effects of individual differences in disorganization, SOA, and prime-target relatedness. If observed semantic priming was associated with automatic or with controlled processes, then the priming effect should be modulated by SOA (yielding an SOA by priming interaction). If the expression of schizotypy in priming was related to either automatic or controlled processes then this SOA by priming interaction should, itself, by modulated by individual differences in (cognitive) disorganization (yielding a disorganization by SOA by priming interaction).

Finally, we reasoned that if schizotypal personality implies illogical semantic storage or further-reaching spreading of semantic activation then we should observe priming effects when prime and target were indirectly related. We predicted that, on average, priming would be reflected in faster responses to targets with related than to targets with unrelated primes but that priming would be greater for directly than for indirectly related primes (yielding an interaction between the effects of relatedness and of the directness of the relation). However, we hypothesized that the association between schizotypy and semantic processing should be evident in both indirect and direct priming conditions for individuals scoring high on disorganization. This implies the prediction that the directness × priming interaction should be smaller (the impact of directness on priming should be reduced) in individuals scoring higher on disorganization, that is, priming should be more similar under direct and indirect prime-target relatedness conditions for individuals scoring higher on disorganization. In the following, we report an investigation in which we manipulated the directness of prime-target relatedness in two sub-studies (direct, indirect priming).

## MATERIALS AND METHODS

### Participants

A group of 81 Psychology students from the University of Barcelona took part in a direct priming sub-study in exchange for course credits (mean age = 19.8, SD = 1.9; 73 females; 69 self-reported as right-handed). A group of 79 different volunteers from the same pool took part in the indirect priming sub-study (mean age = 19.67, SD = 1.75; 71 females; 72 right-handed). They were all native speakers of Spanish with normal or corrected-to-normal vision who reported no history of neurological or psychiatric disease. We gathered written informed consent prior to their participation. The study protocols were approved by the university's ethics committee (lRBOOOO3O99, Comissió de Bioètica de la Universitat de Barcelona, CBUB). All data were recorded anonymously.

### Materials

We selected 60 prime-target pairs from the Normas de Asociación Libre en Castellano database, which provides lists of free association data for different words in Spanish.

For the direct priming sub-study, we selected a related target for each prime from the two most associated words in its list, avoiding phonological overlap between primes and targets. The unrelated prime-target pairs were constructed by pseudorandomly distributing the targets to different primes, while avoiding semantic association and phonological overlap among the unrelated prime-target pairings. For each target word, we created an orthographically plausible pseudoword with the software Wuggy (*Keuleers & Brysbaert, 2010*). The pseudowords matched the letter length, length of subsyllabic segments and transitional frequencies of the real words.

In order to select new targets indirectly associated to the original primes for the indirect priming sub-study, we chose new words from the two most associated words in the directly associated targets' lists of associates. Stimuli that appeared as directly related words in the corresponding prime's list of associates, or words phonologically similar to them, were not included in the final selection. Although, for clarity, we explain stimuli selection for the two sub-studies separately, we selected directly and indirectly related pairs simultaneously.

We measured schizotypal personality by means of two questionnaires: the brief versions of the SPQ (SPQ-B, *Raine & Benishaw, 1995*) and the O-LIFE (sO-LIFE, *Mason, Linney & Claridge, 2005*). We translated these questionnaires into Spanish following common translation and back-translation procedures (*Sierro et al., 2016*). First, a native Spanish speaker of advanced English proficiency translated the English versions into Spanish. Then, an English native bilingual professional translator back-translated the Spanish versions. The minor differences revealed by comparing the original and back-translated versions were discussed by the two translators until agreement. The translated scale items are available at OSF (https://osf.io/j29fn/).

The SPQ-B consists of 22 items assessing cognitive-perceptual deficits (positive dimension, for example, "Have you ever had the sense that some person or force is around you, even though you cannot see anyone?"), interpersonal deficits (negative dimension, for example, "People sometimes find me aloof and distant") and disorganization (disorganized dimension, for example, "People sometimes comment on my unusual mannerisms and habits"). Following *Ferchiou et al. (2017)*, our participants responded to the SPQ-B questionnaire using a 5-point Likert-like scale with one indicating "completely disagree" and five indicating "completely agree". Reliability values for responses of our full participant sample ($n = 160$) were: disorganized, mean = 13.53, SD = 4.71, $\alpha = 0.78$; positive, mean = 17.66, SD = 5.59, $\alpha = 0.75$; negative, mean = 19.89, SD = 6.65, $\alpha = 0.84$. The sO-LIFE consists of 43 yes/no questions tapping into positive traits (e.g., "When in the dark do you often see shapes and forms even though there is nothing there?"), negative traits (e.g., "Are there very few things that you have ever enjoyed doing?"), disorganized traits (e.g., "Are you easily confused if too much happens at the same time?") and impulsive non-conformity traits (e.g., "Do you consider yourself to be pretty much an average sort of person?"—reversed). Reliability values corresponding to our participants' responses to this inventory were: disorganized, mean = 4.49, SD = 2.82, $\alpha = 0.76$; positive, mean = 2.68, SD = 2.1, $\alpha = 0.63$; negative, mean = 2.7, SD = 1.77,

α = 0.62; impulsive nonconformity, mean = 2.39, SD = 1.8, α = 0.54. The participants in the direct and indirect priming sub-studies showed significant differences in their cognitive disorganization scores as measured by SPQ-B (direct priming mean =14.5, SD = 4.8; indirect priming mean =12.6, SD = 4.4; $t(158) = 2.560$, $p = 0.011$), but not by sO-LIFE.

### Design and procedure

We manipulated the directness of the relation between prime and target in two separate sub-studies conducted with different participants. The two studies were identical in procedure but differed in the target stimuli (which were directly or indirectly related to the primes).

In summary, each participant completed 480 lexical decision trials, completing 240 trials responding to word targets and 240 trials to pseudoword targets. In both studies, SOA and relatedness were manipulated within-primes, -targets and-participants. Each participant saw each prime eight times, four times with word targets and four times with pseudoword targets. For the four trials in which a word target was presented, each participant saw the prime two times with one related target (e.g., abdomen-BARRIGA), once at the short SOA and once at the long SOA, and they saw the prime another two times with one unrelated target (e.g., abdomen-MES), once at each SOA. Each person saw each word target four times, twice under the related condition (e.g., abdomen-BARRIGA), once at the short and once at the long SOA, and twice under the unrelated condition (e.g., abeja-BARRIGA), once at the short and once at the long SOA.

For each SOA condition (short or long), we presented the stimuli in two blocks of 120 trials each. Stimuli were presented in different SOA conditions in one of two blocks (short vs. long SOA blocks) to reduce the impact of repetition priming. The order of the blocks and the order of the trials within each block were randomized, and the order of administration of the conditions was counterbalanced across participants. The studies started with four practice trials.

The task was administered using the DMDX software application (*Forster & Forster, 2003*). Stimuli were presented in 4:3 CRT screens with 1,024 × 768 resolution using Arial 14 pt font. Participants were seated 35 cm from the screen. Each trial started with a "+" presented as fixation point for 500 ms. Then, the prime appeared for 200 ms followed by the target word after 50 ms or 550 ms, respectively, for the short or long SOA conditions. The target was onscreen for 200 ms and was followed by a blank screen for 2,000 ms until the next trial began (the response interval) during which participants were able make keypress responses to indicate lexical decisions to targets stimuli (pressing "M" for real words with their right hand and "Z" for pseudowords with their left hand on a QWERTY-type keyboard). The participants were tested in groups of up to four in clone set-ups.

## RESULTS

The full dataset and code for the analyses are available at OSF (https://osf.io/j29fn/).

## Explanation of analysis approach

In our analyses, we used the *brms* library (Bayesian regression models using "Stan"; *Bürkner, 2017*, *2019*; *Carpenter et al., 2017*) to fit Bayesian mixed-effects models. A detailed explanation of our analysis approach is set out in Supplemental Materials (see Article S1) but we summarize the motivation for our approach here. For each effect, we assumed that coefficient estimates may vary in sign and magnitude. Bayesian models are scientifically advantageous because they yield a posterior probability distribution representing the differing probabilities of each potential value of an effect, given the observed evidence and given prior expectations about likely effects. This means that, for each effect, we are able to report the most probable value of the estimate for the effect, while the spread of the posterior distribution directly indicates our uncertainty about the estimate. We report credible intervals (CrI) to summarize that uncertainty.

The study design required the manipulation of prime-target relatedness, the directness of the prime-target relation, and prime-target SOA, in addition to the observation of participants' scores on the SPQ-B and sO-LIFE measures of variation in schizotypy dimensions. We sum-coded (−1, +1) the effects of the categorical variables: prime-target relatedness; SOA; and directness of prime-target relatedness. We standardized participants' scores on the schizotypy dimensions. We fitted separate models including the effects of participant variation on each dimension of one set of schizotypy scales (sO-LIFE or SPQ-B) only. Models were structured to estimate effects of these variables as well as the effects of all interactions up to and including the potential four-way interaction between the effects of directness, schizotypy dimension, SOA and relatedness. We fitted models that included parameters corresponding to random effects associated with:

(1) Unexplained differences between sampled participants in intercepts (random intercepts) and in the within-participant effects of SOA, relatedness and the SOA × relatedness interaction (random slopes); as well as correlations between random intercepts and random slopes;

(2) Unexplained differences between sampled primes or targets in intercepts (random intercepts) and in the within-stimulus effects of participants' variation in schizotypy dimensions, and in the effects of SOA, relatedness and the SOA × relatedness interaction (random slopes); as well as differences in the within-prime effect of directedness; along with correlations between random intercepts and random slopes. (See *Meteyard & Davies, 2020*, for a discussion of the specification of fixed and random effects.)

In the present article, we report the posterior distributions of parameter estimates yielded by models assuming weakly informative priors for fixed effects coefficients or random effects variances: Gaussian (normal) probability distributions centered on a mean of zero with a standard deviation of 10 ($\beta \sim Normal(0, 10)$; $SD \sim Normal(0, 10)$). This assumption of priors expresses the belief that the parameter values would lie between −20 and +20 with 95% probability. To examine the sensitivity of our results to our assumptions, we fitted a series of models with the same fixed and random effects structures but varying prior probability distributions. These sensitivity analyses (see Article S2) indicate that the estimates derived from our models are stable across a range of alternate

assumptions. We also conducted frequentist versions of our analyses which showed similar results (see Article S3) though the most complex models were associated with convergence problems.

## Analysis findings

We report our findings from, firstly, a model including SPQ-B measures of individual differences on schizotypy dimensions, and, secondly, a model including sO-LIFE measures of differences on schizotypy.

### SPQ-B model

We begin by reporting the SPQ-B model effects estimates, together with a quantification of our uncertainty about those estimates, by presenting visualizations of Markov Chain Monte Carlo draws from the posterior distribution of the parameters of the Bayesian model, produced using the *bayesplot* library (*Gabry, 2017*; *Gabry et al., 2019*). Fig. 1 presents density plots computed from posterior draws with all chains merged, with the uncertainty intervals (95% intervals) shown as the shaded areas under the curves. The plots indicate, firstly, that the most probable estimates (located at the vertical lines) for many effects is close to zero. For these effects, including the effects of variation in SPQ-B positive, negative or disorganization dimensions, the impact of individual differences in schizotypy on the latency of correct word classifications was very small. In contrast, we found evidence for effects of SOA, relatedness, and the interaction between relatedness and the directness of prime-target relatedness.

We present a summary of the model in Table 2, showing just the estimates for the effects of the experimental variables and of individual differences on schizotypy dimensions (a full table summary, including fixed and random effects, is presented in Table S1). Every effect is summarized using the mean and the standard error of the posterior distribution together with the upper and lower limits of two-sided 95% credible intervals (lower and upper bounds) based on quantiles (Model summary tables also include Rhat values, a metric of convergence (*Vasishth et al., 2018a*) which should be (and is) close to 1).

Response latencies were about 16 ms longer under the long compared to the short SOA conditions ($\hat{\beta} = 7.76$, 95% CrI [ 4.08–11.42]). (Note that, given the sum coding for the factor—short SOA = −1, long SOA = +1—contrasts are constrained to sum to zero so that the coefficient of the effect corresponds to the difference between the mean latency at one condition level, that is, long SOA, compared to the overall mean across levels, at 0, halfway between short and long SOA.) Response latencies were about 14 ms shorter under the related compared to the unrelated priming conditions ($\hat{\beta} = -7.11$, 95% CrI [−8.85 to −5.35]), with relatedness also sum coded (unrelated = −1, related = +1). Latencies were about 18 ms longer in the direct (coded = 1) compared to the indirect (=−1) condition ($\hat{\beta} = 8.49$, 95% CrI [−0.77 to 17.47]) but the evidence for the effect was relatively weak or uncertain because the breadth of the credible interval encompasses quite broad uncertainty about the magnitude or sign of the effect.

We observed evidence for an interaction between the relatedness and the directness effects. The (negative) effect of prime relatedness was about 5 ms greater under the

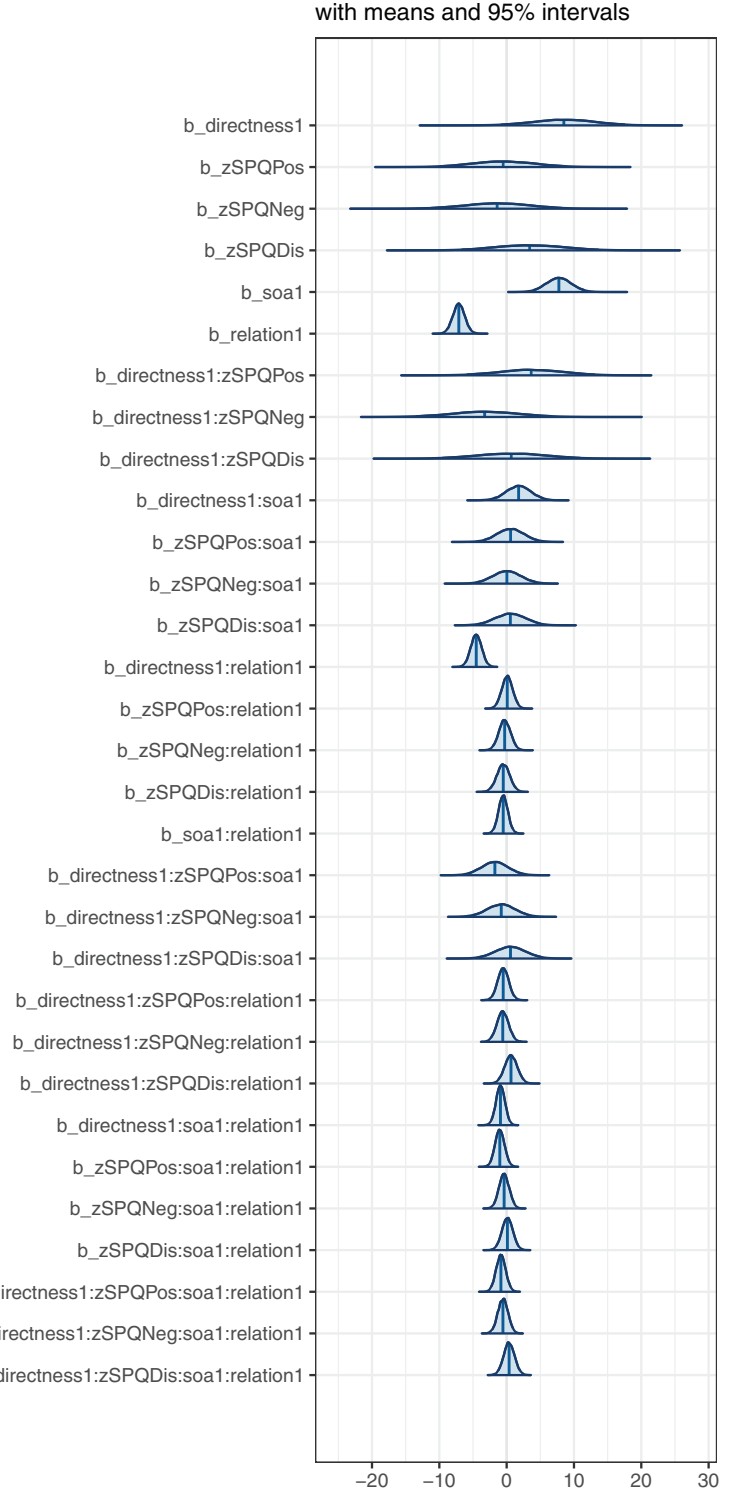

**Figure 1 SPQ analysis—posterior distribution plot for each effect.**

**Table 2 SPQ model summary of population-level effects estimates.**

| Parameter | Mean | SE | Lower bound | Upper bound | Rhat |
|---|---|---|---|---|---|
| Intercept | 615.54 | 5.20 | 605.39 | 625.91 | 1.00 |
| Directness | 8.49 | 4.66 | −0.77 | 17.47 | 1.00 |
| zSPQPos | −0.52 | 4.72 | −9.74 | 8.76 | 1.00 |
| zSPQNeg | −1.41 | 4.95 | −11.23 | 8.16 | 1.00 |
| zSPQDis | 3.41 | 5.16 | −6.60 | 13.60 | 1.00 |
| soa | 7.76 | 1.87 | 4.08 | 11.42 | 1.00 |
| relation | −7.11 | 0.89 | −8.85 | −5.35 | 1.00 |
| Directness:zSPQPos | 3.64 | 4.81 | −5.85 | 13.01 | 1.00 |
| Directness:zSPQNeg | −3.26 | 5.06 | −13.34 | 6.58 | 1.00 |
| Directness:zSPQDis | 0.69 | 5.23 | −9.53 | 10.88 | 1.00 |
| Directness:soa | 1.78 | 1.84 | −1.88 | 5.37 | 1.00 |
| zSPQPos:soa | 0.58 | 2.02 | −3.36 | 4.59 | 1.00 |
| zSPQNeg:soa | 0.04 | 2.11 | −4.12 | 4.20 | 1.00 |
| zSPQDis:soa | 0.56 | 2.27 | −3.94 | 4.96 | 1.00 |
| directness:relation | −4.52 | 0.82 | −6.12 | −2.92 | 1.00 |
| zSPQPos:relation | 0.09 | 0.82 | −1.54 | 1.69 | 1.00 |
| zSPQNeg:relation | −0.29 | 0.88 | −2.01 | 1.42 | 1.00 |
| zSPQDis:relation | −0.50 | 0.95 | −2.39 | 1.36 | 1.00 |
| Soa:relation | −0.51 | 0.70 | −1.90 | 0.88 | 1.00 |
| Directness:zSPQPos:soa | −1.75 | 2.00 | −5.67 | 2.22 | 1.00 |
| Directness:zSPQNeg:soa | −0.80 | 2.13 | −5.05 | 3.41 | 1.00 |
| Directness:zSPQDis:soa | 0.57 | 2.25 | −3.88 | 4.98 | 1.00 |
| Directness:zSPQPos:relation | −0.51 | 0.82 | −2.11 | 1.12 | 1.00 |
| Directness:zSPQNeg:relation | −0.58 | 0.87 | −2.29 | 1.13 | 1.00 |
| Directness:zSPQDis:relation | 0.64 | 0.94 | −1.18 | 2.48 | 1.00 |
| Directness:soa:relation | −0.92 | 0.67 | −2.23 | 0.38 | 1.00 |
| zSPQPos:soa:relation | −1.02 | 0.71 | −2.40 | 0.40 | 1.00 |
| zSPQNeg:soa:relation | −0.38 | 0.76 | −1.87 | 1.12 | 1.00 |
| zSPQDis:soa:relation | 0.11 | 0.82 | −1.50 | 1.72 | 1.00 |
| Directness:zSPQPos:soa:relation | −0.85 | 0.72 | −2.26 | 0.56 | 1.00 |
| Directness:zSPQNeg:soa:relation | −0.55 | 0.76 | −2.06 | 0.93 | 1.00 |
| Directness:zSPQDis:soa:relation | 0.37 | 0.82 | −1.24 | 1.98 | 1.00 |
| Sigma | 49.47 | 0.55 | 48.40 | 50.55 | 1.00 |
| Beta | 141.50 | 0.98 | 139.57 | 143.42 | 1.00 |

direct compared to the indirect priming conditions, given the effect of the interaction ($\hat{\beta} = -4.52$, 95% CrI [−6.12 to −2.92]). Figure 2 illustrates the trends in RT differences associated with this interaction. In contrast, there is little evidence for interactions between SOA and relatedness or between SPQDis differences and relatedness. The points in the plot indicate the mean model fitted outcome, given different values of each predictor variable, conditional on all other predictor variables set to their mean values.
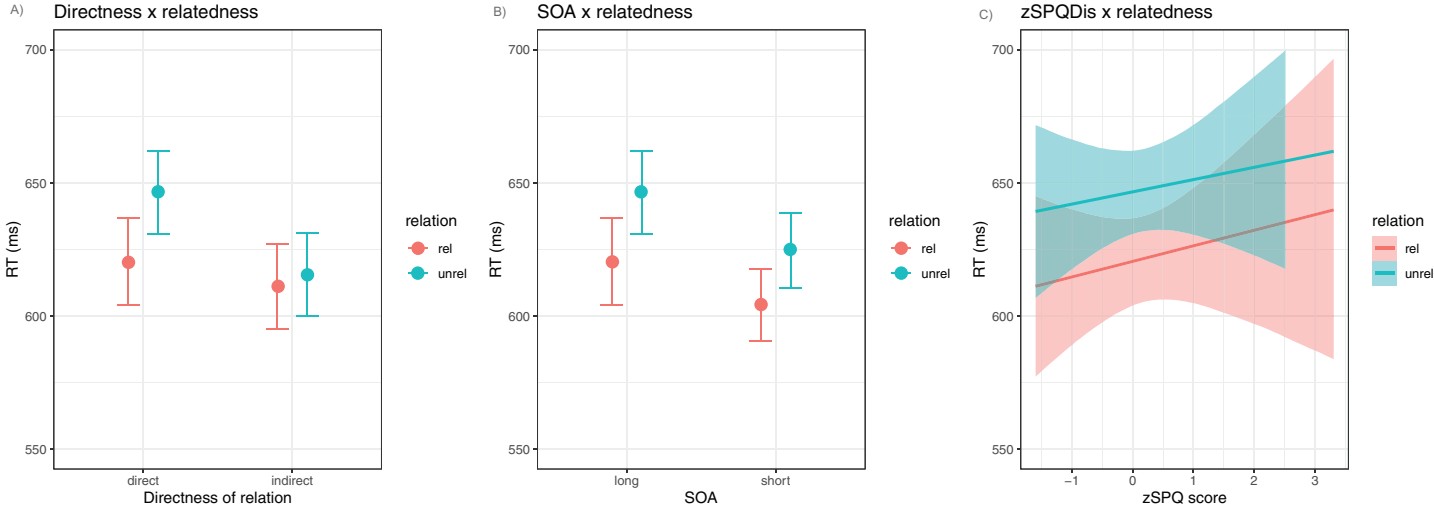

**Figure 2 SPQ analysis—marginal effects plot for the two-way interaction effect.** (A) Directness × relatedness interaction, (B) SOA × relatedness interaction, (C) cognitive disorganization × relatedness interaction.

The error bars indicate the upper and lower bounds of the 95% credible intervals for the estimates.

We hypothesized that individual differences on schizotypy dimensions would modulate: (1) the effect of relatedness, predicting a disorganization by relatedness interaction; (2) the effect of the interaction between SOA and relatedness, predicting a disorganization by SOA by relatedness interaction; and (3) the effect of the interaction between directness and relatedness, predicting a disorganization by directness by relatedness interaction. The posterior distribution plots shown in Fig. 1 clearly indicate how, in contrast to these expectations, the estimates for the effects of the critical interactions were all close to zero.

### sO-LIFE model

The model including sO-LIFE instead of SPQ-B schizotypy dimensions presents a largely similar pattern of effects. For this reason, we present just the tabled summaries of the posterior distribution for parameters (see Table 3), and we invite interested readers to inspect the posterior distribution and marginal effects plots in Supplemental Materials (Figs. S1 and S2) (A full table summary, including fixed and random effects, is presented in Table S2).

What is distinct, compared to the SPQ model, are the indications, firstly, that for at least one schizotypy dimension, OLIFEImp, the most probable effect of variation on that dimension is that response latencies changed perceptibly—RTs increased—albeit by a small amount, in association with increasing scores on the measure ($\hat{\beta} = 5.56$, 95% CrI [−4.30 to 15.29]). The plot presenting the posterior distribution for the effect of OLIFEImp indicates that the estimated effect coefficient is relatively small (under 10 ms), and the uncertainty about the nature of the effect is quite broad, suggesting weak evidence that is, perhaps insufficient to resolve the nature of the trend.

The sO-LIFE model suggests, in addition, that individual differences on some schizotypy dimensions did appear to modulate the effects of the experimental variables but

**Table 3 OLIFE model summary of population-level effects estimates.**

| Parameter | Mean | SE | Lower bound | Upper bound | Rhat |
|---|---|---|---|---|---|
| Intercept | 614.98 | 5.19 | 604.72 | 624.91 | 1.00 |
| Directness1 | 8.27 | 4.64 | −0.87 | 17.32 | 1.00 |
| zOLIFEPos | −0.27 | 4.72 | −9.53 | 9.07 | 1.00 |
| zOLIFENeg | −1.74 | 4.59 | −10.81 | 7.13 | 1.00 |
| zOLIFEDis | 0.84 | 4.89 | −8.73 | 10.42 | 1.00 |
| zOLIFEImp | 5.56 | 4.98 | −4.30 | 15.29 | 1.00 |
| Soa1 | 7.92 | 1.81 | 4.38 | 11.49 | 1.00 |
| Relation1 | −7.30 | 0.87 | −9.02 | −5.58 | 1.00 |
| Directness1:zOLIFEPos | −0.22 | 4.75 | −9.62 | 9.07 | 1.00 |
| Directness1:zOLIFENeg | −1.72 | 4.69 | −10.83 | 7.41 | 1.00 |
| Directness1:zOLIFEDis | 4.52 | 4.96 | −5.20 | 14.22 | 1.00 |
| Directness1:zOLIFEImp | −0.09 | 4.93 | −9.59 | 9.55 | 1.00 |
| Directness1:soa1 | 0.81 | 1.82 | −2.78 | 4.37 | 1.00 |
| zOLIFEPos:soa1 | 0.25 | 1.93 | −3.51 | 4.03 | 1.00 |
| zOLIFENeg:soa1 | 1.22 | 1.88 | −2.45 | 4.86 | 1.00 |
| zOLIFEDis:soa1 | −1.05 | 2.11 | −5.19 | 3.08 | 1.00 |
| zOLIFEImp:soa1 | 5.00 | 2.08 | 0.91 | 9.12 | 1.00 |
| directness1:relation1 | −4.64 | 0.79 | −6.21 | −3.08 | 1.00 |
| zOLIFEPos:relation1 | 0.52 | 0.80 | −1.05 | 2.07 | 1.00 |
| zOLIFENeg:relation1 | −0.45 | 0.77 | −1.96 | 1.07 | 1.00 |
| zOLIFEDis:relation1 | −1.01 | 0.88 | −2.74 | 0.71 | 1.00 |
| zOLIFEImp:relation1 | 0.50 | 0.85 | −1.15 | 2.17 | 1.00 |
| soa1:relation1 | −0.28 | 0.71 | −1.64 | 1.10 | 1.00 |
| directness1:zOLIFEPos:soa1 | 1.06 | 1.91 | −2.71 | 4.84 | 1.00 |
| directness1:zOLIFENeg:soa1 | −0.39 | 1.89 | −4.10 | 3.36 | 1.00 |
| directness1:zOLIFEDis:soa1 | 1.09 | 2.07 | −2.93 | 5.20 | 1.00 |
| directness1:zOLIFEImp:soa1 | −0.87 | 2.05 | −4.91 | 3.18 | 1.00 |
| directness1:zOLIFEPos:relation1 | −0.23 | 0.79 | −1.77 | 1.32 | 1.00 |
| directness1:zOLIFENeg:relation1 | −0.56 | 0.77 | −2.08 | 0.94 | 1.00 |
| directness1:zOLIFEDis:relation1 | −0.89 | 0.87 | −2.58 | 0.82 | 1.00 |
| directness1:zOLIFEImp:relation1 | 2.12 | 0.84 | 0.47 | 3.76 | 1.00 |
| directness1:soa1:relation1 | −0.85 | 0.67 | −2.15 | 0.47 | 1.00 |
| zOLIFEPos:soa1:relation1 | −0.30 | 0.71 | −1.69 | 1.07 | 1.00 |
| zOLIFENeg:soa1:relation1 | −0.43 | 0.69 | −1.80 | 0.92 | 1.00 |
| zOLIFEDis:soa1:relation1 | −0.25 | 0.78 | −1.80 | 1.26 | 1.00 |
| zOLIFEImp:soa1:relation1 | 0.32 | 0.75 | −1.15 | 1.82 | 1.00 |
| Directness1:zOLIFEPos:soa1:relation1 | 0.60 | 0.71 | −0.80 | 2.01 | 1.00 |
| Directness1:zOLIFENeg:soa1:relation1 | −0.79 | 0.69 | −2.15 | 0.56 | 1.00 |
| Directness1:zOLIFEDis:soa1:relation1 | −0.82 | 0.78 | −2.32 | 0.73 | 1.00 |
| Directness1:zOLIFEImp:soa1:relation1 | −0.17 | 0.75 | −1.64 | 1.34 | 1.00 |
| Sigma | 49.41 | 0.55 | 48.32 | 50.49 | 1.00 |
| Beta | 141.45 | 0.98 | 139.52 | 143.39 | 1.00 |

that that influence was observed as subtle, rather small, modulations. Thus, we observed some evidence for potentially interesting interactions. We see some weak evidence for an interaction between the effects of variation in OLIFEDis scores and the impact of directness condition ($\hat{\beta}$ = 4.52, 95% CrI [−5.20 to 14.22]), indicating that for individuals with higher OLIFEDis scores the difference between RTs under direct compared to indirect priming conditions tended to increase. We also see some evidence for an interaction between the effects of variation in OLIFEImp scores and of variation in SOA ($\hat{\beta}$ = 5.00, 95% CrI [0.91–9.12]), indicating that for individuals with higher OLIFEImp scores the difference between RTs under long compared to short SOA conditions tended to increase. Lastly, the model indicated some evidence for a small effect associated with the three-way interaction between the effects of the directness of relation, of variation in OLIFEImp scores, and of relatedness ($\hat{\beta}$ = 2.12, 95% CrI [0.47–3.76]). However, the effect is so small that it can be said to have corresponded to a negligible influence on performance in the lexical decision task.

## DISCUSSION

Our study aimed to ascertain whether individual differences in schizotypal personality, specifically, disorganized traits, are associated with increased spreading of semantic activation. This possibility is of substantive theoretical and practical importance because it could explain symptoms, such as thought disorder, in clinical or sub-clinical populations. Although this hypothesis has been present in the literature for years now, results that have been adduced to support it are far from definitive so that it has not been clear under what circumstances, and in what shape, the association should be observed.

Regarding the impact of a direct semantic relation between lexical stimuli in word recognition tasks, evidence from two previous studies have indicated a significant association between variation in schizotypy and increased priming effects, either under automatic (short SOA) priming conditions (*Neill, Rossell & Kordzadze, 2014*) or under both automatic and controlled priming conditions (*Moritz et al., 1999*). This has led to the claim that individuals scoring high on schizotypy present increased spreading of semantic activation (*Neill, Rossell & Kordzadze, 2014*) compared to individuals scoring low on schizotypy. With respect to evidence of the priming that may result from indirectly related prime-target lexical stimulus pairs, the observation of increased priming effects among individuals scoring high on schizotypy, under both automatic (*Johnston, Rossell & Gleeson, 2008*; *Moritz et al., 1999*) and controlled priming conditions (*Rossell et al., 2014*), has led to the conclusion that individuals with schizotypal traits might present further-reaching spreading of semantic activation (*Johnston, Rossell & Gleeson, 2008*).

Our study does not support any of these hypotheses. We tested a sample of participants ($n$ = 160, in total), with a sample of lexical stimuli (240 word trials per person), considerably larger than that employed in most studies reported in this area. Moreover, we overcame the limitations of previous work by conducting Bayesian mixed-effects models to estimate the effects of the experimental variables (prime-target relatedness, SOA, the directness of the prime-target relation) as well as the effects of individual differences on schizotypal dimensions, and, critically, the effects of interactions between the impact of the

experimental conditions and variation on schizotypy. This approach yielded a high level of sensitivity in our analyses: sufficient for us to be able to identify, with a high degree of certainty, small but distinct effects due to the relatedness of prime-target pairs (a semantic priming effect), the length of SOA, and the modulation of the relatedness effect by the directness of the relation between primes and targets. Indeed, our analyses were sufficient to pick up subtle deviations among participants (estimated as random effects variances) in average reaction time, or in the effects of prime-target relation, of SOA, and of the interaction between relatedness and SOA (see Tables S1 and S2). Impressively, our analyses were sufficient to pick up random variation among responses due to differences between primes in the slopes of the effects of relatedness, directness, SOA or schizotypy variation. These random effect variances were relatively small (SDs of the order of <5) but within the capacity of our methods to derive precise estimates. Yet we observed no influence of cognitive disorganization, and no interaction between the effects of cognitive disorganization and semantic relation.

We acknowledge that it is possible that our participant sample was not sufficient, or not sufficiently representative of the distribution of individual differences in cognitive disorganization in the wider population. However, a comparison of the distribution of our participants' scores with those obtained in previous studies indicate that our sample can be considered to be representative of samples tested in research in the field, potentially, therefore, of the distribution in the general population. Regarding scores in sO-LIFE, the results of our participants covered the full range of possible scores and their distribution characteristics resemble those of the sample assessed in the original scaling study (cognitive disorganization scores in our study: mean = 4.49, SD = 2.82; in *Mason, Linney & Claridge (2005)*: mean = 4.42, SD = 2.9). Regarding scores in the SPQ-B, our results showed slightly lower scores but greater variability than those obtained in scaling studies conducted with an analogous Likert-based version of the test (disorganization scores in our study: mean = 13.53, SD = 4.71; in *Ferchiou et al. (2017)*: mean = 16.77, SD = 1.13). In relation to this, future studies might be interested in assessing a sample comprising both subclinical individuals as well as patients in order to cover the full continuum of thought disorder prevalence.

It should be noted that our analyses indicated a small difference in cognitive disorganization scores as measured by SPQ-B (but not by sO-LIFE) between participants in the direct and indirect priming sub-studies. This difference could be argued to potentially limit the interpretability of the directness effect observed in our research. However, we believe that if a between-group difference in disorganization was confounded with the between-group difference in type of priming (direct vs. indirect) then we should expect the directness effect to be observed in the SPQ-B analysis but not in the sO-LIFE analysis. Instead, we observed some evidence for a directness effect in both analyses though we note that the credible interval suggested considerable uncertainty over the magnitude of the effect.

Our study focused on examining the possible association between variability in semantic processing and differences in thought disorder as measured by cognitive disorganization scales of schizotypy questionnaires in sub-clinical volunteers. However,

some authors have suggested that disturbances in semantic processing could also be associated with individual differences in positive traits such as dissociative experiences (*Dehon, Bastin & Larøi, 2008*; *Winogard, Peluso & Glover, 1998*) hallucinations (*Kanemoto et al., 2013*; *Sugimori, Asai & Tanno, 2011*) or paranormal beliefs (*Meyersburg et al., 2009*; *Pizzagalli, Lehmann & Brugger, 2001*). Our study is also informative in relation to this hypothesis because we employed the questionnaire measures (SPQ-B and sO-LIFE) to estimate individual differences not only in disorganized traits, but also in positive and negative traits. Mirroring the results obtained in relation to disorganized traits, our analyses indicated little or no evidence for interactions between differences in positive symptoms and semantic priming effects.

At least two questions regarding the representativeness of our sample deserve attention. Firstly, around 90% of our participants were females in both sub-studies. Differences in the distribution of schizotypal traits between males and females have been reported in previous work. For instance, *Mason & Claridge (2006)* reported significantly higher cognitive disorganization scores for males compared to females during the creation of O-LIFE. Given the gender imbalance in our sample, our evidence cannot be used to examine whether the interplay between schizotypal traits and semantic processing is affected by gender. Further studies with comparable male and female samples should be conducted to test this possibility.

Another relevant issue concerns the handedness of our participants. Although most self-identified as right-handed, we did not conduct any systematic assessment of their degree of hand preference. Ambilaterality has been associated with schizotypal traits such as magical thinking (*Barnett & Corballis, 2002*), and functional hemispheric asymmetry, which is known to be associated with handedness (*Bourne, 2006*), has been suggested to share a neural basis with schizotypy (*Schmitz et al., 2019*). Along these lines, previous studies have reported stronger priming effects in paranormal believers, compared to disbelievers, only after left visual field presentation of the words (*Mohr, Landis & Brugger, 2006*; *Pizzagalli, Lehmann & Brugger, 2001*). Taking these observations into account, recruitment of homogeneous right-handed samples or samples including comparable groups of right-handed and left-handed participants and left-lateralized presentation of priming stimuli might be more adequate to capture the relation between schizotypal traits and semantic processing and should be considered in further studies.

In our view, the findings we report suggest that previous observations regarding the association between cognitive disorganization and priming should be revisited. As we have already outlined in the introduction, the results in this area present some inconsistency. Yet the potential implications of answers to the research question, whether variation in schizotypy influences the degree or extent of semantic activation, seem to us to merit further research in this area. Our findings indicate that that further research will require the completion of studies that are sufficiently powered to permit the identification of what are likely to be subtle interactions.

One limitation of our study is related to the use of self-report schizotypal questionnaires to measure cognitive disorganization. These kinds of assessment tools have been predominant in the literature studying the relation between schizotypy and semantic

priming effects. In our work, we used the short versions of the two questionnaires most commonly used in previous studies, focusing on the disorganization dimensions. In the SPQ-B, this dimension includes items tapping into odd speech (directly related to thought disorder) and odd behavior. Regarding sO-LIFE, the analogous dimension of cognitive disorganization includes items specifically assessing thought disorder, but also other traits such as poor attention and decision-making or social anxiety. Taking this into account, the cognitive disorganization scales of these two questionnaires might not be specific enough to precisely assess thought disorder, not to mention, to capture the usually overlooked heterogeneity of this phenomenon (*Sass & Parnas, 2017*). In this sense, more objective measures of this trait, such as word graph analyses (*Mota et al., 2012*; *Mota, Copelli & Ribeiro, 2017*) or repetition and predictability indexes (*Linscott, 2005*) might be more adequate to capture the possible association between schizotypy and variability in semantic processing in future studies.

## CONCLUSIONS

In sum, our study provides no compelling evidence that schizotypal traits, specifically those associated with the cognitive disorganization dimension, reflect enhanced semantic processing, as evidenced by variation in semantic activation in priming tasks. Our findings warrant the conclusion that the effects of interest may be relatively small, and thus require high powered investigations for precise estimation in future work.

## ACKNOWLEDGEMENTS

All analyses were run using The High End Computing (HEC) facility at Lancaster University. We gratefully acknowledge the support of Mike Pacey in using the HEC to complete our analyses.

### Funding

This study was supported by the grants PSI2016-80061-R (AEI/FEDER, UE) from the Agencia Estatal de Investigación of the Spanish government and the European Regional Development Fund, and grant 2017SGR387 (AGAUR) from the Catalan government, both to Javier Rodríguez-Ferreiro. The funders had no role in study design, data collection and analysis, decision to publish, or preparation of the manuscript.

### Grant Disclosures

The following grant information was disclosed by the authors:
Agencia Estatal de Investigación of the Spanish government and the European Regional Development Fund (AEI/FEDER, UE): PSI2016-80061-R.
AGAUR: 2017SGR387.

### Competing Interests

The authors declare that they have no competing interests.

## Author Contributions

- Javier Rodríguez-Ferreiro conceived and designed the experiments, performed the experiments, analyzed the data, prepared figures and/or tables, authored or reviewed drafts of the paper, and approved the final draft.
- Mari Aguilera performed the experiments, authored or reviewed drafts of the paper, and approved the final draft.
- Rob Davies analyzed the data, prepared figures and/or tables, authored or reviewed drafts of the paper, and approved the final draft.

## Human Ethics

The following information was supplied relating to ethical approvals (i.e., approving body and any reference numbers):

The study protocols were approved by the university's ethics committee (lRBOOOO3O99, Comissió de Bioètica de la Universitat de Barcelona, CBUB).

## Data Availability

The full dataset and code for the analyses are available at OSF: https://osf.io/j29fn/.

## Supplemental Information

Supplemental information for this article can be found online at http://dx.doi.org/10.7717/peerj.9511#supplemental-information.

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
