# Peer review of "Semantic priming and schizotypal personality: reassessing the link between thought disorder and enhanced spreading of semantic activation"

_PeerJ, doi:10.7717/peerj.9511_

## Round 0.1 · original submission · Major Revisions

I have received very thoughtful reviews from Philip Sumner (Reviewer 2) and another expert in the field. Both are quite positive about the manuscript but see room for improvement. Given the clarity of their feedback, I have little to add here. However, one theme that emerges from both reviews is the need for greater theoretical grounding in the manuscript, addressing the basis for the hypothesized relationship between schizotypy and semantic relatedness in predicting rates of lexical decision. I agree that the paper would benefit from the addition of more theory.

Beyond this, I request that you add a statement to the paper confirming whether, for all experiments, you have reported all measures, conditions, data exclusions, and how you determined your sample sizes. You should, of course, add any additional text to ensure the statement is accurate. This is the standard reviewer disclosure request endorsed by the Center for Open Science [see http://osf.io/project/hadz3]. I include it in every review.

I look forward to reading a revision of this work!

Reviewer 1 ·

Basic reporting

Please see attached PDF.

Experimental design

Please see attached PDF.

Validity of the findings

Please see attached PDF.

Additional comments

Please see attached PDF.

Annotated reviews are not available for download in order to protect the identity of reviewers who chose to remain anonymous.

·

Basic reporting

Writing style is generally comprehensible and professional, though expression is somewhat awkward or repetitive in places. Some of the presented information might gain clarity if the writing was more succinct, as sometimes the relevance of the information being presented is difficult to determine through the density of details provided.

Within the introduction, a strong statistical argument is made to account for the mixed findings from previous investigations of semantic priming and schizotypy. This argument is made from an appropriate coverage of the previous literature. Furthermore, the authors introduce some relevant theoretical and methodological concepts, such as semantic networks and the various ways in which the semantic priming paradigm can be manipulated to emphasise different aspects of the functioning of these networks. They also aptly consider the severity and spread of sampled thought and language disorder by comparing them the normative data in their discussion. Thus, relevant prior literature is referenced.

On the other hand, there are some gaps in the introduction, as well as in the discussion. Firstly, the only link made between schizophrenia and schizotypy was that there is a similarity between the personality traits of schizotypy and the symptoms of schizophrenia (lines 39-46). No clear theoretical argument was presented which alludes to the reasons why people with more disorganized schizotypal personalities should necessarily be expected to share similar semantic abnormalities as people with schizophrenia who demonstrate symptoms of thought and language disorder, if shared mechanisms are expected at all. Indeed, the authors avoid bringing in literature surrounding the semantic correlates of thought and language disorder reported in samples of people with schizophrenia (an area of research which is also arguably associated with mixed findings). Admittedly, direct comparisons between schizotypy and schizophrenia are beyond the scope of the study, but the rationale for exploring semantic function and schizotypy needs elaboration. A brief statement explicitly denoting the theoretical continuity between schizotypy traits and the symptoms of schizophrenia might therefore be helpful to round out the introduction. Meehl’s model of schizotaxia, schizotypy and schizophrenia is an example of one influential model that somewhat emphasises disorganization (Meehl, 1962, Am Psychol, v17, pp827; see also Lenzenweger, Maher & Manschreck, 2005, J Clin Psychol, v61, pp1295). The relationship between schizotypy and schizophrenia should not merely be implied because shared aetio-pathogenic mechanisms may not follow from shared phenomenology (Waters & Fernyhough, 2019, Schiz Bull, v45, pp717).

Secondly, although many methodological details were introduced comprehensively, the relevance of some of these details to the study rationale were not entirely clear. This was most prominent in relation to the schizotypy measures (lines 75-93), where the measurement scales and subscales were outlined without an explanation as the theoretical significance of these differences, or why both were required in the current study. The O-LIFE cognitive disorganization scale arguably has a broader granularity than the SPQ incoherent speech scale because it includes items that assess attention and concentration, decision-making, and social anxiety (Mason & Claridge, 2006, Schiz Res, v82, pp203). Hence, I imagine the SPQ is more closely aligned to the research aims of the study, which concerns thought disorder specifically. I note that the findings reported from one of the previous studies suggests a divergence between the semantic correlates of the SPQ and the Frankfurt Compliant Questionnaire (lines 115-119), and differences in the granularity of constructs being measured may be a factor in such differences. Furthermore, thought and language disorders are a heterogeneous group of symptoms that can be represented by a number of factors, including evidence of multidimensionality when self-reported thought disorder is measured in non-clinical samples using dedicated thought disorder questionnaires (Barrera et al., 2015, Int J Psychol and Psychological Therapy, v15, pp155). At the same time, some researchers have expressed scepticism towards the degree of cognitive disorganization that can be adequately assessed via self-report measures. Thus, in the discussion, how well the short-versions of the SPQ disorganization and OLIFE cognitive disorganization scales represent schizotypal thought and language disorder should be carefully considered.

Thirdly, given that the research aim specifically emphasises thought and language disorders/cognitive disorganization, the findings of previous semantic priming studies (lines 111-130) should be re-organized slightly to better distinguish the correlates of schizotypal thought and language traits from the correlates of schizotypy more generally. This reflects the findings of semantic research into schizophrenia, where the general correlates of schizophrenia have been suggested to differ somewhat from those specifically of thought disorder (e.g. Doughty & Done, 2009, Cog Neuropsychiatry, v14, pp473). Indeed, the cognitive correlates might differ amongst the different aspects of thought disorder (e.g. Bora et al., 2019, Schiz Res, v209, pp2).

The manuscript, as well as the accompanying tables and figures, conform broadly to the professional article structure, and the study is self-contained with results relevant to hypotheses.

Experimental design

The work fits within the scope of the journal. This is an empirical study of psychological processes. As such, the work is contained within the field of the Social Sciences, but with implications for the Health Sciences.

The rationale of the study was clear: to replicate previous semantic priming studies with more sensitive methodologies. This is a relevant and meaningful aim. However, given the heterogeneity in both findings and methodology that characterises research into the semantic mechanisms of schizophrenia and of schizotypy, it would also seem prudent to state clearly the theory driving the hypotheses in this study. Some minor tweaking is possible in this regard, particularly with regards to the precise theory guiding each hypothesis. For instance, Spitzer (1997, Schiz Bull, v23, pp29) proposed a model where thought disorder arises from an increased distance in the spreading of automatic semantic activation across nodes in a network. This increased spread of activation results in the facilitation of more distantly related concepts, which can be measures as an increased semantic priming effect for indirectly related words at low SOAs. The theory does not predict a relationship between thought disorder and the magnitude of direct semantic priming, or for priming at longer SOAs. The authors did hypothesise a reduction in the difference in semantic priming effect between directly and indirectly related lexical pairs with greater cognitive disorganization scores (lines 270-275), which would be predicted by Spitzer’s model. However, the direction of the influence of SOA upon the relationship between the semantic priming effect and cognitive disorganization scale scores was not specified, but was presented as a research question (lines 255-264). Which theory of thought disorder was used to predict differences at longer SOAs?

The work seems to have been conducted rigorously and reported in great detail. Translation of questionnaires was conducted with back-translation. Analysis decisions seem to have been carefully considered, explicitly stated, justified with references, and presented with a high degree of transparency. The results seem to be stable across the several methodological variants presented. Disorganization scores associated with reasonable internal consistencies. Reproducibility is facilitated through the use of automated software packages and libraries (presumably in R). Furthermore, many supplementary materials have been provided, such as a link to the translated questionnaires, documented analysis assumptions and inclusion of the raw data.

Validity of the findings

This study represents meaningful replication with a revised analysis approach in an effort to overcome the poor statistical power and sensitivity of previous studies. The discussion
and conclusions are conservative, but are linked to the original hypotheses and do not overextend from the data.

The authors acknowledge the between-group comparisons required for the directness condition (lines 445-450), and that these factors were accounted for (lines 452-453). However, the authors might consider clearly indicating to the readers in more general terms whether the two subsamples differed in their disorganization scores, and whether the directness condition was confounded by any between-group differences.

One of the weaknesses of the findings may arguably stem from the particular self-report measures used, for reasons mentioned already (see Basic Reporting section). The authors could consider discussing alternative and more objective measures of schizotypal thought and language traits for use in future research.

Additional comments

The authors present an interesting study in an area of research that is showing increased interest. The overall study rationale is admirable; to replicate the semantic priming paradigm that has been used previously to investigate schizotypy, but with more sensitive analytic methods in an effort to address the poor statistical power that has characterised these past studies. The methodological rigour is therefore a relative strength of the current study, although the theoretical context surrounding the study should be improved upon.

---

## Round 0.2 · Minor Revisions

I thank you for your thoughtful revision. Both of the original reviewers have read through the fresh manuscript, and both are positive about its contribution to the literature. However, both see some areas that could be improved. Their comments appear below.

I concur with Reviewer 1, who provides advice on a subtle revision to the final sentence of your Abstract that will make it more readable for a broad audience. I encourage you to follow their advice.

Reviewer 2 (Philip Sumner) was satisfied with the changes to the content of your manuscript, but has new concerns about the delivery of this content. I echo Dr. Sumner's sentiment. In a revision, I would like to see you provide a more organized synthesis of the literature. I would also like you to follow his advice on shortening the Results section to better highlight the primary analyses that relate to your hypotheses. While the level of detail you've provided is laudable, it does reduce the readability of the manuscript. You may choose to include some of this additional detail in Supplementary Materials.

If you can address these stylistic issues, I see no reason that the manuscript will not be acceptable for publication at PeerJ.

Reviewer 1 ·

Basic reporting

This is the review of a (first) revision, so I am brief and write everything I have to say here:

The authors did a laudable task in integration my (and my peer reviewer's) comments, thank you. I am entirely satisfied with all the changes in the MS and the authors' rebuttal. In the case they renounced to elaborate on the suggestions (e.g. handedness / lateralization issue), they justify why, and I accept this. (I do not want to be picky, but just as a feedback to the authors: you need not lateralize stimuli to get lateralized processing: given the (claimed) fact that remote and indirect associations are more a domain of the right hemisphere, the issue of (handedness-associated) hemispheric language specialization remains relevant to research that compares direct and indirect priming and I would suggest in such cases to have a uniform right-handed population or two handedness groups of comparable size).

OK with me to keep the topic of false memories.

The only thing which could maybe be changed is one sentence in the Abstract: it now reads: "Our study provides no compelling evidence that schizotypal symptoms, specifically those associated with the cognitive disorganization dimension, are rooted in enhanced semantic processing, as reflected by increased spreading of semantic activation in priming tasks completed by sub-clinical individuals."
The phrase "enhanced semantic processing" is difficult to understand by somebody who îs naive to the topic of semantics and priming. So, it might be more clear to write:
"Our study provides no compelling evidence that schizotypal symptoms, specifically those associated with the cognitive disorganization dimension, are rooted in an
increased spreading of semantic activation in priming tasks."
(The "completed by sub-clinical individuals" can be omitted to have the sentence shorter and given the context of the Abstract).

Editor: please decide on yourself whether you think it might help readability.

Experimental design

(responded to in first round)

Validity of the findings

(responded to in first round)

Additional comments

The authors did a laudable task in integration my (and my peer reviewer's) comments, thank you. I am entirely satisfied with all the changes in the MS and the authors' rebuttal. In the case they renounced to elaborate on the suggestions (e.g. handedness / lateralization issue), they justify why, and I accept this. (I do not want to be picky, but just as a feedback to the authors: you need not lateralize stimuli to get lateralized processing: given the (claimed) fact that remote and indirect associations are more a domain of the right hemisphere, the issue of (handedness-associated) hemispheric language specialization remains relevant to research that compares direct and indirect priming and I would suggest in such cases to have a uniform right-handed population or two handedness groups of comparable size).

OK with me to keep the topic of false memories.

The only thing which could maybe be changed is one sentence in the Abstract: it now reads: "Our study provides no compelling evidence that schizotypal symptoms, specifically those associated with the cognitive disorganization dimension, are rooted in enhanced semantic processing, as reflected by increased spreading of semantic activation in priming tasks completed by sub-clinical individuals."
The phrase "enhanced semantic processing" is difficult to understand by somebody who îs naive to the topic of semantics and priming. So, it might be more clear to write:
"Our study provides no compelling evidence that schizotypal symptoms, specifically those associated with the cognitive disorganization dimension, are rooted in an
increased spreading of semantic activation in priming tasks."
(The "completed by sub-clinical individuals" can be omitted to have the sentence shorter and given the context of the Abstract).

·

Basic reporting

The theoretical justification for the study has now been bolstered with the introduction of Meehl’s model to indicate (albeit briefly) the possible relationship between schizophrenia and schizotypy, and Spitzer’s model to indicate that thought disorder may arise from an increased distance of spreading of activation throughout semantic networks. The consideration of additional limitations and important avenues for future research has also strengthened the discussion. My only remaining critiques pertain to 1) the clarity and concision of writing/expression in the introduction; and 2) the clarity of the depiction of the main findings relevant to the hypotheses in the results section. Accordingly, the introduction and results sections are probably excessive in length.

1. Although the overall writing style and tone is appropriate, the construction of sentences and linkages between sentences can be somewhat convoluted and repetitive in places, particularly in the first half of the introduction. The removal of statements such as "as we explain following" (line 274) and breaking down some of the lengthier sentences might help. In addition, a greater synthesis of the literature in some sections may help emphasise the arguments being made. A specific example and suggestions are offered for your consideration:

Lines 169-192: This is one example where the writing is replete with details, making it harder to identify the specific contention and the relevant lines of evidence. The writing might be more succinct with a greater level of synthesis. Currently in this section, the findings and methodology from past research is described study-by-study. Instead, the previous findings could be further organized around the models for which they demonstrate support (or lack thereof). Spitzer argued that indirect priming best captured any increased spreading of semantic activation that might underpin thought disorder. Thus, the findings from indirect priming with a short SOA should be presented together. By contrast, direct priming presumably probes the degree of 'appropriate' semantic access, so the findings for direct priming should be presented separately. In addition, other methodological variability is evident from the details provided here (e.g. comparing groups defined based on the median versus correlation analyses). These methodological details do not need to be reported until later, when the argument is made that the sensitivity of prior studies has been influenced their analysis approach. I note that the authors do provide a critical summative reflection of the literature (showing a good level of synthesis) later in the introduction (lines 209-224) – however, these paragraphs could be combined to make the work more concise.


2. The transparency and detail of the analysis conducted is respectable. All the required information is provided. However, given the length of the results section, some readers will feel overwhelmed when trying to extract the main findings. Many of the main effects are described in detail (e.g. lines 558-568 pertain to the influence of SOA on response times). Is it possible to emphasise the main findings that are relevant to the hypotheses (i.e. the interactions that include schizotypy) as opposed to the main effects for the various conditions alone? Similarly, Figures 2 and 3 clearly indicate the presence of semantic priming in the SPQ model, with Figure 3 showing an interaction between the magnitude of this priming and the directness of the semantic relationship. However, the modulation of these relationships with schizotypy is not depicted as clearly.
Line 638: What OLIFE subscale is being referred to here?


Other minor comments that are not crucial but the authors may wish to consider:

Line 66-76: This addition of the possible positive aspects of schizotypy is an interesting and important idea. The current argument links thought disorder to semantic activation, and semantic activation to creativity. Is there any research that links thought disorder, particularly non-clinical trait-based thought disorder, with creativity?

Line 53: To avoid confusion, some distinction should be made here between the thought disorder described and specific disorders of thought content, such as delusions.

Lines 720: The authors acknowledge that their sample may not capture the full distribution of cognitive disorganization, though their sample indeed matched the normative sample for the OLIFE. This could be an opportunity to mention the broader continuum model of schizotypy and schizophrenia, and the possibility of using cross-diagnostic samples in the future to more faithfully capture the underlying processes (see Roche et al., 2015, Schiz Bull, v 41, pp 951 for information of the continuum of the prevalence and severity of thought disorder).

Lines 779-791: This is a nice discussion of the use of objective measures, particularly quantitative measures based on speech samples, as a potentially more sensitive assessment of schizotypal disorganization. The breadth of phenomena encompassed in the OLIFE measure of cognitive disorganization was also mentioned. The authors may want to note that clinical thought disorder itself is multidimensional and can be heterogeneous across individuals. It is not clear whether or not schizotypal thought disorder/cognitive disorgaization is similarly multidimensional and whether using a global disorganization score best captures the underlying construct.

Experimental design

Nothing further to add.

Validity of the findings

Raw data is accessible. Nothing further to add.

Additional comments

The work contained within the manuscript remains extensive and shows a high degree of transparency. The authors have implemented most of the suggested revisions - they have introduced more theory to ground their findings, and elaborated on some avenues for future research. Thus, I feel that the theoretical rationale for the study and validity of outcomes are now at an appropriate level. However, I have suggested some additional changes that are limited mostly to the introduction and results sections, and pertain mostly to the expression and organization of the information presented.

---

## Round 0.3 · accepted · Accept

I am satisfied that all of the critiques from reviewers have been addressed in this revision, and the reorganization of the piece has increased its readability. Bravo. This is an important and interesting contribution to the literature.